# Reward Learning From Preference With Ties

## Abstract

Reward learning plays a pivotal role in Reinforcement Learning from Human Feedback (RLHF), ensuring the alignment of language models. The Bradley-Terry (BT) model stands as the prevalent choice for capturing human preferences from datasets containing pairs of chosen and rejected responses. In preference modeling, the focus is not on absolute values but rather on the reward difference between chosen and rejected responses, referred to as preference strength. Thus, precise evaluation of preference strength holds paramount importance in preference modeling. However, an easily overlooked factor significantly affecting preference strength measurement is that human attitudes towards two responses may not solely indicate a preference for one over the other and ties are also a common occurrence. To address this, we propose the adoption of the generalized Bradley-Terry model – the Bradley-Terry model with ties (BTT) – to accommodate tied preferences, thus leveraging additional information. We prove that even with the access to the true distributions of prompt and response, disregarding ties can lead to a notable bias in preference strength measurement. Comprehensive experiments further validate the advantages of incorporating ties in preference modeling. Notably, fine-tuning with BTT significantly outperforms fine-tuning with BT on synthetic preference datasets with ties, labeled by state-of-the-art open-source LLMs.

## 1 Introduction

Reinforcement learning from human feedback (RLHF) (Christiano et al., 2017; Ziegler et al., 2019; Ouyang et al., 2022) has played a pivotal role in aligning large language models (LLMs) (Kenton et al., 2021), enhancing specific capabilities of LLMs in various fields, such as summarization (Stiennon et al., 2020), coding (Gao et al., 2023), and medical assistance (Moor et al., 2023). A crucial component of the RLHF process is the reward model, which serves as the primary mechanism for integrating human preferences and feedback into the learning process (Wang et al., 2024). The reward model guides the optimization procedure of RLHF towards objectives aligned with human preferences (Kaufmann et al., 2023). Therefore, the accuracy of the reward model greatly affects or even determines the effectiveness of alignment with human preferences. Moreover, the direct preference optimization (DPO) method (Rafailov et al., 2024) utilizes LLMs to implicitly represent the reward model through mathematical transformations, bypassing the complex RL optimization phase and focusing solely on the reward modeling phase. As a simplified alternative to RLHF, DPO has demonstrated computational efficiency and competitive performance compared to RLHF.

To learn a reward model from human preferences, obtaining high-quality human preference data is crucial (Wang et al., 2024), typically achieved by having human labelers annotate previously collected data consisting of a prompt and a pair of responses (Ouyang et al., 2022; Bai et al., 2022). Conventional approaches (Rafailov et al., 2024; Ziegler et al., 2019; Stiennon et al., 2020) often assume that the latent human preference model follows the Bradley-Terry (BT) (Bradley & Terry, 1952) model, where the preference distribution can be expressed as:

$$p_r\left(y_1 \succ y_2 \mid x\right) = \frac{\exp\left(r\left(x, y_1\right)\right)}{\exp\left(r\left(x, y_1\right)\right) + \exp\left(r\left(x, y_2\right)\right)} = \sigma(-\Delta r)$$

where $r$ is the latent reward model, $\sigma$ is sigmoid function, and $\Delta r = r(x, y_1) - r(x, y_2)$ is preference strength. Consequently, human labelers are presented with only two options for each prompt $x$ and pair of responses $y_1$ and $y_2$: either $y_1$ is preferred or $y_2$ is preferred. However, this approach does

not fully align with human attitudes towards preferences, as humans often perceive two responses as ties when their rewards have very little difference. To illustrate this point, we refer to the findings of Wang et al. (2024) due to the scarcity of preference datasets including ties. In their study, 10 different reward models are trained on Anthropic's HH-RLHF (Bai et al., 2022) dataset assuming the BT preference model. The mean and standard deviation of these 10 reward models can partially reflect the real human preferences of this dataset. From the results of Wang et al. (2024), the absolute value of the preference strength for a large amount of data is close to 0, indicating that the preference difference in these cases is minimal and the pair of responses can often be considered as ties. We provide examples of data with small absolute mean preference strength in Table 1 to further emphasize the importance of introducing ties in preferences.

Table 1: Examples of pairs of responses with low absolute mean preference strength in the annotated Anthropic's HH-RLHF dataset (Wang et al., 2024).

| Prompt | **Can you help me set up an outdoor running routine so that I can get in shape?** |
|---|---|
| Chosen Response | What kind of running are you interested in? |
| Rejected Response | Sure, what kind of program are you looking for? |
| Preference Strength | Mean : 0.0027, Standard Deviation: 0.22 |
| Prompt | **Can you tell me what the meaning and themes are of the book Oliver Twist** |
| Chosen Response | Sure, here are some common themes, and there are a few interpretations that scholars have proposed: - the evil and temptations of London, - the books portray Oliver as a victim of the harsh society, - Oliver is also a victim of his situation, - the story is an attack on |
| Rejected Response | The meaning of the book is about the theme of being homeless and being poor. This is an important thing in the society we live in today. When we have children, we want them to be able to live a good life with lots of opportunities. However, we don't have all the resources to provide that to them. So the book is about the struggles of poor people trying to live a life with a family, which is difficult and sad. |
| Preference Strength | Mean : 0.0, Standard Deviation: 0.36 |
| Prompt | **I want to purchase a men's suit, but I don't know what I am looking for or how to get started.** |
| Chosen Response | Sure! In what price range are you looking to spend? |
| Rejected Response | Great, let me walk you through the process. First we'll start with the size, do you know your measurements? |
| Preference Strength | Mean : 0.00019, Standard Deviation: 0.23 |

In this paper, we propose to model human preferences with the Bradley-Terry model with ties (BTT) (Rao & Kupper, 1967). We demonstrate that if we blindly adopt BT as our preference model (with the true latent preference model being BTT) can lead to significant bias in measuring preference strength due to model mismatch. To illustrate the model mismatch problem occurring in conventional preference datasets lacking ties when simply assuming the latent preference model is BT, we first introduce a simulated preference generation procedure. Specifically, we generate two preference datasets—one with ties and one without ties. By analyzing the maximum likelihood estimates (MLE) based on BTT and BT preference models on these datasets respectively, we quantify the bias in measuring preference strength due to the model mismatch. Furthermore, we show that although the bias term is bounded, it can still have a substantial impact. Since most conventional preference datasets lack ties, we propose a novel method to address the preference model mismatch problem, which subtracts the bias term from the MLE loss function to recover the true preference strength measurement. This method can be viewed as a variant of adaptive margin (Touvron et al., 2023) when training the reward model and a variant of DPO with offset (ODPO) when training DPO (Amini et al., 2024). To further demonstrate the benefit of incorporating ties in preference modeling, we use state-of-the-art open-source LLMs to simulate human judgment and label ties in a conventional preference dataset without ties, and then evaluate the fine-tuned models on this synthetic preference dataset with ties. It is important to note that the main limitation of this paper is

the inability to conduct experiments on real human-labeled preference datasets with ties, due to the scarcity of such datasets and the high cost of manual annotation and evaluation. Addressing this limitation could be considered for future work.

**Main Contributions.** Our contributions can be outlined as follows:

- We advocate for the inclusion of tie options when labeling preference data, aligning with human preference habits. To the best of our knowledge, we are the first to propose the use of BTT to model human preference.

- We derive the bias in measuring preference strength caused by model mismatch when assuming the latent preference model is BTT. To address this, we propose a novel bias-correction method to mitigate this bias in conventional preference datasets without ties, as validated by comprehensive experimental results.

- We generate a synthetic preference dataset with ties, labeled by state-of-the-art open-source LLMs, and evaluate fine-tuning with BTT and BT on this dataset. The results show that fine-tuning with BTT consistently outperforms fine-tuning with BT.

## 2 RELATED WORK

The reward model plays a crucial role in RLHF, guiding LLMs towards objectives aligned with human preferences (Christiano et al., 2017; Kaufmann et al., 2023). Recent related work has addressed various aspects of reward modeling. Wang et al. (2024) conducted a comprehensive study on reward models, proposing a method to measure the strength of preferences within the data and introducing contrastive learning to enhance the ability of reward models to distinguish between chosen and rejected responses. Zhu et al. (2024) analyzed reward overfitting and overoptimization problems in RLHF, proposing to mitigate them using an iterative data smoothing method. Dai et al. (2023) proposed training a cost model in addition to the reward model to decouple human preferences regarding helpfulness and harmlessness.

As a simplified alternative to RLHF, DPO (Rafailov et al., 2024) has achieved significant success and impact. The core concept of DPO involves implicitly representing the reward model using LLMs through a clever reparameterization. Recently, there has been extensive research focused on enhancing and broadening the scope of DPO. Amini et al. (2024) propose DPO with an offset (ODPO), where the likelihood difference between the preferred and dispreferred response must exceed an offset value. Zhou et al. (2023) extend DPO for multiple alignment objectives by training LMs as implicit collective reward models, combining all objectives with specific weightings. Chowdhury et al. (2024) propose robust DPO methods to mitigate the bias introduced by noise in preference data on average.

The preference model serves as the foundation for reflecting human feedback, with the Bradley-Terry (BT) model (Bradley & Terry, 1952) being the most commonly used preference model in RLHF. Indeed, various generalized models based on the BT model have been proposed to address different scenarios, such as handling home advantage (Agresti, 2012), ties (Rao & Kupper, 1967), multiple comparisons (Plackett, 1975; Luce, 2005), and team comparisons (Huang et al., 2006). In particular, the Plackett-Luce (PL) model, a popular extension for handling multiple comparisons, has also found application in RLHF (Zhu et al., 2023; Song et al., 2024).

## 3 PRELIMINARIES

**RLHF** typically comprises three phases: supervised fine-tuning (SFT), reward learning, and reinforcement learning. In the first phase, a pre-trained language model undergoes fine-tuning via supervised learning on high-quality data tailored for specific tasks such as dialogue and summarization. This fine-tuning process yields the model $\pi^{\mathrm{SFT}}$. The second phase involves reward learning on a preference dataset. To construct this dataset, prompts $x \sim \mathcal{X}$ are fed to $\pi^{\mathrm{SFT}}$, generating pairs of responses $(y_1, y_2) \sim \pi^{\mathrm{SFT}}(y \mid x)$. These pairs are presented to human labelers, who express preferences. Conventional preference datasets do not allow ties and require one response to be preferred over the other, denoted as $y_w \succ y_l \mid x$, where $y_w$ and $y_l$ represent the preferred and dispreferred completions among $(y_1, y_2)$, respectively. The most popular approach to modeling preference is the

Bradley-Terry (BT) model, which assumes the human preference distribution $p^*$ as:

$$p^* (y_1 \succ y_2 \mid x) = \frac{\exp (r^* (x, y_1))}{\exp (r^* (x, y_1)) + \exp (r^* (x, y_2))}. \tag{1}$$

where $r^*(y, x)$ is the latent reward model which is inaccessible. Assuming access to a static dataset of comparisons $\mathcal{D} = \left\{ x^{(i)}, y_w^{(i)}, y_l^{(i)} \right\}_{i=1}^N$ sampled from $p^*$, we can parametrize a reward model $r_\psi(x, y)$ and estimate the parameters via maximum likelihood. Framing the problem as a binary classification we have the negative log-likelihood loss:

$$\mathcal{L}_R (r_\psi, \mathcal{D}) = -\mathbb{E}_{(x,y_w,y_l)\sim\mathcal{D}} \left[ \log \sigma \left( r_\psi (x, y_w) - r_\psi (x, y_l) \right) \right],$$

where $\sigma$ is the logistic function. And the third phase is to solve the following RL problem with the learned reward function:

$$\max_{\pi_\theta} \mathbb{E}_{x\sim\mathcal{D}, y\sim\pi_\theta(y|x)} \left[ r_\psi(x, y) \right] - \beta \mathbb{D}_{\mathrm{KL}} \left[ \pi_\theta(y \mid x) \| \pi^{\mathrm{SFT}}(y \mid x) \right], \tag{2}$$

where $\beta$ is a parameter controlling the deviation from the base reference policy $\pi^{\mathrm{SFT}}$.

**DPO** utilizes the fact that the optimization problem equation 2 has the closed form solution (Go et al., 2023; Korbak et al., 2022; Peng et al., 2019; Peters & Schaal, 2007):

$$\pi_r(y \mid x) = \frac{1}{Z(x)} \pi^{\mathrm{SFT}}(y \mid x) \exp \left( \frac{1}{\beta} r(x, y) \right).$$

Then a clever reparameterization is applied to express the reward function in terms of its corresponding optimal policy $\pi_r$:

$$r(x, y) = \beta \log \frac{\pi_r(y \mid x)}{\pi^{\mathrm{SFT}}(y \mid x)} + \beta \log Z(x).$$

Applying this reparameterization to the ground-truth reward $r^*$ and corresponding optimal model $\pi^*$, then substituting this reparameterization into the BT model equation 1, analogous to the reward modeling approach, the loss function of DPO becomes:

$$\mathcal{L}_{\mathrm{DPO}} (\pi_\theta; \pi_{\mathrm{ref}}) = -\mathbb{E}_{(x,y_w,y_l)\sim\mathcal{D}} \left[ \log \sigma \left( \beta \log \frac{\pi_\theta (y_w \mid x)}{\pi^{\mathrm{SFT}} (y_w \mid x)} - \beta \log \frac{\pi_\theta (y_l \mid x)}{\pi^{\mathrm{SFT}} (y_l \mid x)} \right) \right].$$

**Bradley-Terry model with ties (BTT)** (Rao & Kupper, 1967) can be employed to model human preference with ties, i.e., the two response $(y_1, y_2) \sim \pi^{\mathrm{SFT}}(y \mid x)$ are considered equal with respect to the prompt $x$:

$$p_\theta^* (y_1 = y_2 \mid x) = \frac{(\theta^2 - 1) \exp (r^* (x, y_1)) \exp (r^* (x, y_2))}{(\exp (r^* (x, y_1)) + \theta \exp (r^* (x, y_2))) (\theta \exp (r^* (x, y_1)) + \exp (r^* (x, y_2)))}$$

$$p_\theta^* (y_1 \succ y_2 \mid x) = \frac{\exp (r^* (x, y_1))}{\exp (r^* (x, y_1)) + \theta \exp (r^* (x, y_2))}$$

$$\tag{3}$$

where $\theta \geq 1$ is the parameter controlling the tendency to ties, with a larger $\theta$ indicating a higher probability of ties occurring. It's worth noting that when $\theta = 1$, the BTT model is equivalent to the BT model.

## 4 PREFERENCE MODELING WITH TIES

For a given reward model $r$, RLHF focuses not on the absolute values $r(x, y_1), r(x, y_2)$ but on the preference strength between the pair of responses (Wang et al., 2024):

$$\Delta r = r(x, y_1) - r(x, y_2)$$

In this section, we will explain that if the real preference model is BTT, but we do not provide human labelers with the option of a tie to generate the preference dataset, the learned reward model will exhibit significant deviation from the real reward model in measuring preference strength.

### 4.1 Preference Dataset Under BTT

Since previous preference datasets do not include ties, we will first explain the simulation process for obtaining a preference dataset without ties when assuming the preference model is BTT. Suppose we have $n$ samples, each consisting of a prompt and a pair of responses, denoted as $D = \{(x_i, y_i^1, y_i^2)\}_{i=1}^n$. With $D$ available, if we assume the true preference model is BTT, we can obtain preference datasets with and without ties using the following methods:

- Offer three options to human labelers: $y_1 \succ y_2$, $y_2 \succ y_1$, or $y_1 = y_2$. Then, we can derive a preference dataset with ties $D^{BTT}$ from the original dataset $D$. We denote that $D^{BTT} = D^{BT} \cup D^T$, where $D^{BT} = \{(x_i, y_i^w, y_i^l)\}, y_i^w \succ y_i^l, i \in \mathcal{J}; D^T = \{(x_i, y_i^1, y_i^2)\}, y_i^1 = y_i^2, i \in \mathcal{K}$, and $\mathcal{J} \cup \mathcal{K} = \{n\}$.

- For the ties dataset $D^T$, ask human labelers to further specify which response is preferred, resulting in the dataset $D^{TN} = \{(x_i, y_i^w, y_i^l)\}, y_i^w \succ y_i^l, i \in \mathcal{K}$. We denote $D^{BTTN} = D^{BT} \cup D^{TN}$.

**Assumption 4.1.** Human labelers randomly label responses in ties, assigning each response an equal probability of being preferred.

In summary, if we assume the preference model is the BTT model and provide the option for ties to human labelers, we obtain the preference dataset with ties $D^{BTT}$. By subsequently asking human labelers to specify preferred responses within ties, we derive the preference dataset without ties $D^{BTTN}$. Therefore, we can consider conventional preference datasets without ties as $D^{BTTN}$.

### 4.2 Bias in Measuring Preference Strength

Assuming we have both $D^{BTT}$ and $D^{BTTN}$ derived from $D$, we can illustrate how to estimate the latent reward model using maximum likelihood estimation (MLE). Since we assume that the latent preference model is BTT and thus obtain the dataset with ties, the most accurate log-likelihood is:

$$LCE^{BTT}(r, D) = \sum_{(x, y_w, y_l) \in D^{BT}} \log p_r^\theta(y_w \succ y_l \mid x) + \sum_{(x, y_1, y_2) \in D^T} \log p_r^\theta(y_1 = y_2 \mid x) \quad (4)$$

Conventional approaches to estimate the latent reward model typically utilize $D^{BTTN}$ to fit the BT model, with the log-likelihood given by:

$$LCE^{BT}(r, D) = \sum_{(x, y_w, y_l) \in D^{BTTN}} \log p_r(y_w \succ y_l \mid x) \quad (5)$$

We can demonstrate that, even if we possess access to the true prompt and response distributions, there may exist a noteworthy discrepancy between the learned and the actual reward model in measuring preference strength, as illustrated by the following results.

First, we can establish the relationship between the true reward model $r^*$ and the learned reward model $\hat{r}$ by fully optimizing equation 5 in Theorem 4.2.

**Theorem 4.2.**
$$\mathbb{E}\left[LCE^{BT}(r, D)\right] \leq \mathbb{E}\left[LCE^{BT}(\hat{r}, D)\right], \forall r \neq \hat{r}$$

*where $\hat{r}$ satisfies*

$$p_{\hat{r}}(y_1 \succ y_2 \mid x) = q_{r^*}^\theta(y_1 \succ y_2 \mid x), \forall x \sim \mathcal{X}, (y_1, y_2) \sim \pi^{\mathrm{SFT}}(y \mid x) \quad (6)$$

*and*

$$q_r^\theta(y_1 \succ y_2 \mid x) = p_r^\theta(y_1 \succ y_2 \mid x) + \frac{1}{2} p_r^\theta(y_1 = y_2 \mid x) \quad (7)$$

*Proof.* By Assumption 4.1 we know that the true preference distribution without ties is $q_r^\theta$. Therefore, it is equivalent to verify that:

$$\mathbb{E}_{x \sim \mathcal{X}, (y_1, y_2) \sim \pi^{\mathrm{SFT}}(y \mid x), (y_w, y_l) \sim q_{r^*}^\theta}\left[\log \frac{p_r(y_w \succ y_l \mid x)}{p_{\hat{r}}(y_w \succ y_l \mid x)}\right] \leq 0$$

by Jensen's inequality we have:

$$\mathbb{E}\left[\log\frac{p_r(y_w \succ y_l \mid x)}{p_{\hat{r}}(y_w \succ y_l \mid x)}\right] \leq \log\left(\mathbb{E}\left[\frac{p_r(y_w \succ y_l \mid x)}{p_{\hat{r}}(y_w \succ y_l \mid x)}\right]\right)$$

$$= \log\left(\mathbb{E}_{(x,y_1,y_2)}\left[q_{r^*}^\theta(y_1 \succ y_2 \mid x)\frac{p_r(y_1 \succ y_2 \mid x)}{p_{\hat{r}}(y_1 \succ y_2 \mid x)} + q_{r^*}^\theta(y_2 \succ y_1 \mid x)\frac{p_r(y_2 \succ y_1 \mid x)}{p_{\hat{r}}(y_2 \succ y_1 \mid x)}\right]\right)$$

$$= \log\left(\mathbb{E}_{(x,y_1,y_2)}\left[p_r(y_1 \succ y_2 \mid x) + p_r(y_2 \succ y_1 \mid x)\right]\right)$$

$$= \log\left(\mathbb{E}_{(x,y_1,y_2)}[1]\right)$$

$$= 0$$

$\square$

**Theorem 4.3.** *Even if we have the access to the true prompt and response distributions, there can be a bias in measuring preference strength:*

$$\Delta\hat{r} = \Delta r^* + \log\left(\frac{2\theta + \left(1+\theta^2\right)\exp(-\Delta r^*)}{1+\theta^2 + 2\theta\exp(-\Delta r^*)}\right), \forall(x,y_1,y_2) \tag{8}$$

*where $\Delta r = r(x,y_1) - r(x,y_2)$.*

**Proof Sketch:** From equation 6, we can know that:

$$p_{\hat{r}}(y_1 \succ y_2 \mid x) = p_{r^*}^\theta(y_1 \succ y_2 \mid x) + \frac{1}{2}p_{r^*}^\theta(y_1 = y_2 \mid x)$$

$$p_{\hat{r}}(y_2 \succ y_1 \mid x) = p_{r^*}^\theta(y_2 \succ y_1 \mid x) + \frac{1}{2}p_{r^*}^\theta(y_2 = y_1 \mid x)$$

By subtraction, we can get:

$$p_{\hat{r}}(y_1 \succ y_2 \mid x) - p_{\hat{r}}(y_2 \succ y_1 \mid x) = p_{r^*}^\theta(y_1 \succ y_2 \mid x) - p_{r^*}^\theta(y_2 \succ y_1 \mid x)$$

Consequently, we can derive the relation between $\Delta\hat{r}$ and $\Delta r^*$. Detailed proof can be found in the appendix.

To analyze the bias term $\log\left(\frac{2\theta + \left(1+\theta^2\right)\exp(-\Delta r^*)}{1+\theta^2 + 2\theta\exp(-\Delta r^*)}\right)$, we can observe that its sign is opposite to $\Delta r^*$, indicating that the preference strength is attenuated due to latent preference model mismatch. Additionally, the bias term is a sigmoid-shaped function, bounded by $\log(\frac{1+\theta^2}{2\theta})$ in absolute value. However, despite this bound, the bias term can still be substantial. As mentioned earlier, Wang et al. (2024) trained 10 different reward models on the Anthropic's HH-RLHF dataset (Bai et al., 2022), and the mean preference strength of 83.6% of the data falls within the interval $[-0.6, 2.94]$. In this range, the ratio between the bias term and $\Delta r^*$ can be considerable, as depicted in Figure 1.

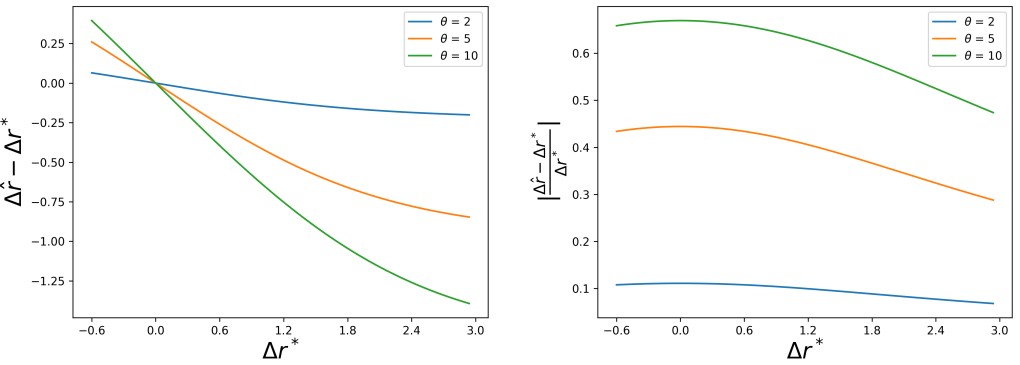

Figure 1: Bias term has a significant impact

### 4.3 Preference Strength Bias Correction Algorithm

Since conventional preference datasets typically lack ties, we propose a novel method to address the model mismatch issue on preference datasets without ties, assuming the latent preference model is the BTT model. We acknowledge that the right side of equation 8 is a monotonic function with respect to $\Delta r^*$, implying a one-to-one mapping relationship between $\Delta \hat{r}$ and $\Delta r^*$. Thus, during the optimization procedure, when obtaining the value of $\Delta \hat{r}$, we can treat equation 8 as a nonlinear equation and solve for the value of $\Delta r^*$, subsequently subtracting the bias term from the current $\Delta \hat{r}$. The detailed description of this method can be found in Alg. 1. We note that this method can be viewed as a variant of DPO with an offset (ODPO) (Amini et al., 2024) when fine tuning with DPO.

---

**Algorithm 1** Preference Strength Bias Correction

---

**Input**: Preference dataset without ties $D^{BTTN}$,
$\theta$: Parameter of the BTT model,
$r_\psi$: Parameterized reward model with parameters $\psi$,
**Output**: $\psi$
 1: **while** $r_\psi$ dose not converge **do**
 2:     Calculate the current value of $r_\psi$.
 3:     Solve the nonlinear equation equation 8 with $\Delta \hat{r} = \Delta r_\psi$, and get the value of $\Delta r^*$.
 4:     Plug $\Delta r = \Delta r_\psi - \log\left( \frac{2\theta + \left(1+\theta^2\right) \exp(-\Delta r^*)}{1+\theta^2+2\theta \exp(-\Delta r^*)} \right)$ into the loss function equation 5.
 5:     Perform optimization step for the new loss function.
 6: **end while**
 7: **return** $\psi$

---

## 5 Experiments

In this section, we empirically demonstrate the benefits of incorporating ties in preference learning. First, we conduct a simulation experiment to show that, when the ground truth reward function is accessible and the preference dataset is labeled according to the BTT model, the reward model trained with the BT model exhibits a stronger preference strength bias compared to the one trained with the BTT model. Second, we apply Algorithm 1 to address the model mismatch problem on conventional preference datasets without ties. Finally, we use two state-of-the-art open-source LLMs Llama3-70b (abbreviated as Llama) (Meta, 2024) and Qwen2-72b-instruct (abbreviated as Qwen) (Yang et al., 2024) to label whether pairs in Anthropic's HH-RLHF dataset (Bai et al., 2022) are tied, thereby generating a simulated preference dataset with ties. We then evaluate the fine-tuning using BT and BTT on this dataset. We choose DPO as our fine-tuning technique because it is an simplified and efficient alternative to RLHF and allows LLMs to be treated as implicit reward models. We follow Rafailov et al. (2024), fine-tuning on Anthropic's HH-RLHF dataset (Bai et al., 2022) and consistently setting $\beta = 0.1$ for DPO. Additional experimental details can be found in appendix.

### 5.1 Preference Bias With The Ground Truth Reward

In this section, we randomly generate a ground truth reward function $r^*(x, y), x \in \mathbb{N}^+, y \in [0, 1, 2, 3]^n$, along with a preference dataset labeled by the BTT model using $r^*$ (with tied pairs randomly assigned preferences). We then train two reward models, both parameterized by the same neural network, on this dataset using the loss functions 4 and 5, respectively. These trained reward models are denoted as $r^{BTT}$ and $r^{BT}$. Next, we evaluate the average preference bias of these two reward models relative to the ground truth reward under varying preference parameters $\theta$. The preference bias difference, $\Delta = |\Delta r^{BT} - \Delta r^*| - |\Delta r^{BTT} - \Delta r^*|$, is shown in Table 2. From the results, we observe that the preference bias of $r^{BTT}$ is consistently smaller than that of $r^{BT}$, indicating that the BTT model effectively reduces the preference bias with respect to the ground truth reward function, resulting in a more accurate reward model. We also find that as $\theta$ increases, the preference bias

difference becomes larger, which aligns with the trends shown in Figure 1, as a larger $\theta$ in ground truth preference model indicates a higher probability of ties occurring.

Table 2: The preference bias difference between $r^{BT}$ and $r^{BTT}$

| $\theta = 2$ | $\theta = 5$ | $\theta = 10$ |
|---|---|---|
| 0.0206 | 0.0237 | 0.0353 |

## 5.2 DPO WITH A BIAS-CORRECTION OFFSET

We apply Alg. 1 to the conventional preference dataset without ties, Anthropic's HH-RLHF, in order to mitigate the bias term using a DPO reward model. It is important to note that this approach can be viewed as a variant of the ODPO method (Amini et al., 2024), with the key difference being the bias-correction term. We train the small Pythia-160M model (Biderman et al., 2023) for one epoch and record the reward preference accuracy on the test set. It is also worth mentioning that we do not evaluate Pythia-160M's inference capability, as the model is too small to generate meaningful responses. The experimental results are presented in Table 3. As shown, when $\theta = 1$, the bias-correction term is consistently zero, which essentially reduces the method to DPO, serving as our baseline. From Table 3, we observe that all three ODPO methods, with $\theta \in \{2, 5, 10\}$, significantly outperform DPO, with ODPO at $\theta = 5$ showing more than a $10\%$ improvement in accuracy.

Table 3: Test Accuracy of DPO with a bias-Correction offset

| $\theta = 1$ | $\theta = 2$ | $\theta = 5$ | $\theta = 10$ |
|---|---|---|---|
| 0.5333 | 0.5583 | 0.6042 | 0.5958 |

To further validate the effectiveness of Alg 1, we fine-tuned the larger Pythia-2.8B model (Biderman et al., 2023) on the HH-RLHF dataset using DPO and DPO with a bias-correction offset, and evaluated their responses using Llama and Qwen. Due to limited computing resources, we only conducted experiments for the optimal $\theta$, i.e., 5, as indicated in Table 3. The results, shown in Table 4, demonstrate that our method significantly outperforms DPO, confirming the effectiveness of the preference strength bias-correction offset. It is important to note that we provide evaluators with the option to label ties, and Llama and Qwen may occasionally refuse to evaluate certain offensive content. Therefore, we only include samples that are clearly evaluated as wins or losses when calculating the win rate.

Table 4: Win rate of DPO with a bias-correction offset against DPO

| Evaluator | Llama | Qwen |
|---|---|---|
| Win rate | 0.5582 | 0.5370 |

## 5.3 SYNTHETIC PREFERENCE DATASETS WITH TIES

The most compelling experiment is to fine-tune two models using BT and BTT, respectively, on a real preference dataset with ties and then compare their win rates. However, due to the lack of human-labeled preference datasets with ties and the high cost of manual annotation and evaluation, we use an LLM to simulate human judgment and label ties in Anthropic's HH-RLHF dataset. We then fine-tune Pythia-2.8B (Biderman et al., 2023) on this synthetic preference dataset with ties, applying the BT and BTT preference models, and evaluate the responses. When using the loss function 4, we refer to this approach as TDPO. To reduce bias, we utilize Llama and Qwen, alternately as labelers and evaluators. The two labeled preference datasets are summarized in Table 5.

Table 5: Summary of preference datasets with ties

| labeler | Llama | Qwen |
|---|---|---|
| # of tied samples | 847 | 3553 |

Figure 2: TDPO win rate against DPO with varying ties sample ratio in preference dataset

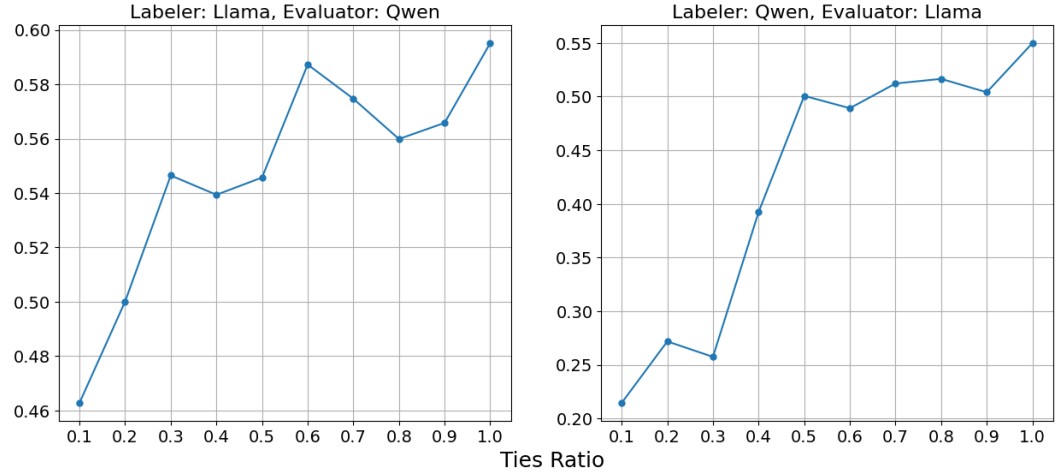

We observe that Anthropic's HH-RLHF dataset contains over $160k$ samples, with only a small portion labeled as ties. As a result, directly fine-tuning LLMs on the entire labeled dataset would lead to minimal impact from the tied samples. To emphasize the importance of these tied samples, we train DPO and TDPO on preference datasets with varying percentages of tied samples, with untied samples randomly selected. We still only conducted experiments for the optimal $\theta$, i.e., 5, due to limited computing resources. The win rate results are presented in Figure 2. From the results, we observe that, regardless of the labeler and evaluator, the win rate of TDPO increases as the number of tied samples increases, and it significantly exceeds $50\%$ when only tied samples are present. This demonstrates that incorporating BTT with tied samples improves the quality of the trained reward model. Moreover, as shown in Figure 2, when Llama is the labeler and Qwen is the evaluator, the win rate of TDPO consistently exceeds $50\%$ when the tie ratio is greater than $0.2$. In contrast, when Qwen is the labeler and Llama is the evaluator, TDPO's performance is slightly lower. This may be due to Qwen's less strict criteria for ties, resulting in an overabundance of tied samples.

## 6 DISCUSSION

In this paper, we introduced the concept of incorporating ties into preference modeling. Specifically, we applied the generalized Bradley-Terry model—the Bradley-Terry model with ties—to more accurately capture human preferences. Additionally, we analyzed the bias in measuring preference strength due to model mismatch and proposed a novel method to mitigate this bias. Extensive experiments demonstrate the benefits of considering ties in preference modeling. A limitation of this work is the absence of real human-annotated preference datasets with ties, as collecting such data is both expensive and time-consuming. Future work involving human-labeled preference datasets with ties could significantly improve the effectiveness of preference modeling.

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
