Therefore,

$$\frac{\exp(\Delta\hat{r})}{1 + \exp(\Delta\hat{r})} - \frac{1}{1 + \exp(\Delta\hat{r})} = \frac{\exp(\Delta r^*)}{\theta + \exp(\Delta r^*)} - \frac{1}{1 + \theta\exp(\Delta r^*)}$$

$$\frac{\exp(\Delta\hat{r} - \Delta r^*)}{1 + \exp(\Delta\hat{r})} - \frac{1}{(1 + \exp(\Delta\hat{r}))\exp(\Delta r^*)} = \frac{1}{\theta + \exp(\Delta r^*)} - \frac{1}{(1 + \theta\exp(\Delta r^*))\exp(\Delta r^*)}$$

Then we can get:

$$\exp(\Delta\hat{r} - \Delta r^*)$$

$$= (1 + \exp(\Delta\hat{r}))\left[\frac{1}{\theta + \exp(\Delta r^*)} - \frac{1}{(1 + \theta\exp(\Delta r^*))\exp(\Delta r^*)} + \frac{1}{(1 + \exp(\Delta\hat{r}))\exp(\Delta r^*)}\right]$$

$$= \frac{1 + \exp(\Delta\hat{r})}{\theta + \exp(\Delta r^*)} + \frac{\theta\exp(\Delta r^*) - \exp(\Delta\hat{r})}{(1 + \theta\exp(\Delta r^*))\exp(\Delta r^*)}$$

$$= \frac{1 + \exp(\Delta\hat{r})}{\theta + \exp(\Delta r^*)} + \frac{\theta - \exp(\Delta\hat{r} - \Delta r^*)}{1 + \theta\exp(\Delta r^*)}$$

Consequently,

$$\exp(\Delta\hat{r} - \Delta r^*)\frac{2 + \theta\exp(\Delta r^*)}{1 + \theta\exp(\Delta r^*)} = \frac{1 + \exp(\Delta\hat{r})}{\theta + \exp(\Delta r^*)} + \frac{\theta}{1 + \theta\exp(\Delta r^*)}$$

Then,

$$\exp(\Delta\hat{r} - \Delta r^*) = \frac{1 + \theta\exp(\Delta r^*)}{2 + \theta\exp(\Delta r^*)} \cdot \frac{1 + \exp(\Delta\hat{r})}{\theta + \exp(\Delta r^*)} + \frac{\theta}{2 + \theta\exp(\Delta r^*)}$$

$$= \frac{1 + \theta\exp(\Delta r^*)}{2 + \theta\exp(\Delta r^*)} \cdot \frac{\exp(-\Delta r^*) + \exp(\Delta\hat{r} - \Delta r^*)}{1 + \theta\exp(-\Delta r^*)} + \frac{\theta}{2 + \theta\exp(\Delta r^*)}$$

$$\exp(\Delta\hat{r} - \Delta r^*)\left(1 - \frac{1 + \theta\exp(\Delta r^*)}{(2 + \theta\exp(\Delta r^*))(1 + \theta\exp(-\Delta r^*))}\right)$$

$$= \exp(\Delta\hat{r} - \Delta r^*)\frac{1 + \theta^2 + 2\theta\exp(-\Delta r^*)}{(2 + \theta\exp(\Delta r^*))(1 + \theta\exp(-\Delta r^*))}$$

$$= \frac{(1 + \theta\exp(\Delta r^*))\exp(-\Delta r^*)}{(2 + \theta\exp(\Delta r^*))(1 + \theta\exp(-\Delta r^*))} + \frac{\theta}{2 + \theta\exp(\Delta r^*)}$$

$$= \frac{\theta + \exp(-\Delta r^*)}{(2 + \theta\exp(\Delta r^*))(1 + \theta\exp(-\Delta r^*))} + \frac{\theta}{2 + \theta\exp(\Delta r^*)}$$

Finally, we can get:

$$\exp(\Delta\hat{r} - \Delta r^*) = \frac{\theta + \exp(-\Delta r^*)}{1 + \theta^2 + 2\theta\exp(-\Delta r^*)} + \frac{\theta(1 + \theta\exp(-\Delta r^*))}{1 + \theta^2 + 2\theta\exp(-\Delta r^*)}$$

$$= \frac{2\theta + (1 + \theta^2)\exp(-\Delta r^*)}{1 + \theta^2 + 2\theta\exp(-\Delta r^*)}$$

Denote

$$f(x) = \frac{2\theta + (1 + \theta^2)\exp(x)}{1 + \theta^2 + 2\theta\exp(x)}$$

then we know that:

$$f'(x) = \frac{\exp(x)\left(1 - \theta^2\right)^2}{\left(1 + \theta^2 + 2\theta\exp(x)\right)^2} \geq 0$$

Therefore,

$$\frac{2\theta}{1 + \theta^2} = \lim_{x \to -\infty} f(x) \leq f(x) \leq \lim_{x \to \infty} f(x) = \frac{1 + \theta^2}{2\theta}$$

Consequently, we have:

$$|\Delta\hat{r} - \Delta r^*| \leq \log(\frac{1 + \theta^2}{2\theta})$$

$\square$

## B EXPERIMENT DETAILS

### B.1 EXPERIMENTAL SETUP

For each single experiment, we choose the same $64$ batch size, RMSprop optimizer, $1e - 5$ learning rate and $\beta = 0.1$. All experiments are conducted on 4 Nvidia A800-80GB GPUs in a single node.

### B.2 WIN RATE PROMPT FOR LLAMA AND QWEN

We use the same prompt for Llama and Qwen to evaluate a pair of responses:

*For the following query to a chatbot, which response is more helpful?*
*Query: [ ]*
*Response A: [ ]*
*Response B: [ ]*
*FIRST provide a one-sentence comparison of the two responses and explain which you feel is more helpful. SECOND, on a new line, state only "A", "B", "Neither" or "Both" to indicate which response is more helpful. Your response should use the format: Comparison: [one-sentence comparison and explanation] More helpful: ["A", "B", "Neither" or "Both"]*

### B.3 PROMPT FOR LLAMA AND QWEN TO LABEL TIES

We use the same prompt for Llama and Qwen to label whether a pair of responses are tied:

*For the following query to a chatbot, are the two responses equally good?*
*Query: []*
*Response A: []*
*Response B: []*
*Answer with exactly "Yes" or "No".*