# OpenReview forum: "Reward Learning From Preference With Ties"
_ICLR.cc/2025/Conference — Submitted to ICLR 2025_

### Official Review · Reviewer_VzFu · 2024-11-02

**Soundness:** 2
**Presentation:** 1
**Contribution:** 2
**Rating:** 3
**Confidence:** 4

**Summary:**

This paper introduces the Bradley-Terry model with ties (BTT) to improve the accuracy of human preference modeling in reinforcement learning, addressing the bias introduced by ignoring ties. It demonstrates through theoretical analysis and experimental validation that BTT outperforms the traditional BT model, especially in datasets with ties, thereby enhancing the alignment of large language models with human preferences.

**Strengths:**

1. **Relevance of Preferences with Ties**: The inclusion of ties in preference modeling is highly significant, as real-world scenarios often involve ranking difficulties. This makes the problem both relevant and worthy of exploration.

2. **Clarity and Readability**: The paper is well-structured and easy to follow. The logical flow and clear explanations enhance the overall readability, making the content accessible and understandable.

**Weaknesses:**

1. **Poor Layout and Presentation**: The formatting of figures (e.g., Figure 1, Figure 2) and tables (e.g., Table 2, Table 3, Table 4, Table 5) is suboptimal, often appearing overly large and occupying excessive space. This layout issue detracts from the readability and professional appearance of the paper, suggesting a rushed submission. Enhancing the visual presentation and ensuring that figures and tables are appropriately sized would significantly improve the reader's experience.

2. **Lack of Baseline Comparisons**: While the paper introduces a novel method, it fails to compare its performance against established baselines. Specifically, the proposed method is described as a variant of DPO with an offset (ODPO) (Amini et al., 2024), but no direct comparisons are provided. This omission makes it difficult to assess the effectiveness and reliability of the new approach. Including such comparisons would strengthen the paper's claims and provide more concrete evidence of its contributions.

3. **Limited Dataset Diversity**: The experiments rely solely on the HH-RLHF dataset, which limits the generalizability of the findings. Using a more diverse set of datasets would help validate the robustness and broader applicability of the proposed method. Expanding the experimental scope to include additional datasets would provide a more comprehensive evaluation.

**Questions:**

The current work appears rushed, both in experimental design and writing. To improve, the authors should optimize figure and table layouts for better readability, include comparisons with baseline methods to validate the effectiveness of the proposed BTT model, and diversify the datasets used to ensure the generalizability of the findings. A more thorough and polished approach is needed.

---

### Official Review · Reviewer_AidY · 2024-11-03

**Soundness:** 2
**Presentation:** 3
**Contribution:** 2
**Rating:** 3
**Confidence:** 3

**Summary:**

This paper investigates the impact of annotated ties in preference data on subsequent reward differences and subtask fine-tuning. The authors generalize the Bradley-Terry model to include ties (BTT) and demonstrate the advantages of incorporating ties in preference modeling from both theoretical and experimental perspectives.

**Strengths:**

- Provides theoretical insights on the importance of incorporating ties in preference modeling
- Proposes an algorithm to address model mismatch problems in conventional preference datasets that lack tie annotations
- Conducts comprehensive experiments to verify the proposed methodology

**Weaknesses:**

Importance of the problem:
- While incorporating ties in preference modeling is conceptually important, questions arise about its practical significance.  If ties occur infrequently in real annotations, as noted in lines 457-459 ("We observe that Anthropic's HH-RLHF dataset contains over 160k samples, with only a small portion labeled as ties"), the significance of this issue may be limited.
- Moreover, how does the significance of ties change with data scaling? Would the impact of ties diminish as the dataset size increases?

Questions about experiments:
- How are the parameter   of the BTT model $\theta$ determined?
- While the experiments were conducted on synthetic preference datasets, would it be possible to do experiments using publicly available real-annotated datasets?

**Questions:**

Please refer to weakness part.

---

### Official Review · Reviewer_k4NC · 2024-11-04

**Soundness:** 2
**Presentation:** 2
**Contribution:** 2
**Rating:** 3
**Confidence:** 3

**Summary:**

This paper investegates the limitation of current RLHF and preference optimization approaches in LLMs: current approaches use the Bradley-Terry (BT) model which forces binary choices between two options, but doesn't account for when responses might be equally good (ties). To address this, the authors propose to using the Bradley-Terry model with ties (BTT) instead of the standard BT to better reflect human preference patterns with ties (TDPO). Afterwards, the experiments demonstrate improved performance using synthetic datasets with ties labeled by open-source LLMs.

**Strengths:**

It is interesting to notice ties might exist in current preference dataset, which might bring up some issue in current preference optimization approaches.

The paper is easy to follow.

**Weaknesses:**

I do not see the necessity to introduce algorithm to deal with ties from Table 1. As stated in Lines 56-58, the reward models are trained with BT assumption. And Table 1 shows that these reward models could not distinguish the ties in the datasets due to 0 preference strength. However, the chosen and rejected responses in Table 1 are both suitable as the preferred response. The reward models of BT assign similar rewards to the chosen and rejected responses, which means the BT reward model could already recognize ties without explicitly introducing BTT. Therefore, the necessity of introducing BTT should be further explained.

The experiment is not sufficient enough. Only one dataset (HH) is used with only one baseline model (DPO). Preference optimization has been explored for more than a year. There are several datasets that could be used to evaluate the performance of preference optimization. Moreover, more recent advances in preference optimization, especially those considering  noise in datasets, should be reviewed and be integrated as baselines.

**Questions:**

How to infer the necessity of explicitly introducing ties from Table 1?

---

### Official Review · Reviewer_RCHs · 2024-11-04

**Soundness:** 2
**Presentation:** 3
**Contribution:** 2
**Rating:** 3
**Confidence:** 4

**Summary:**

This paper investigates the importance of handling ties in RLHF. The traditional Bradley-Terry (BT) model for preference modeling assumes that there is always one option that is clearly preferred, without considering the fact that humans may consider some options as ties. To address this issue, the authors propose using an extended Bradley-Terry model (BTT), which can more accurately reflect human preferences. They show that when the actual preference model should include ties but there are no such annotations in the dataset, it will lead to bias in the measurement of preference. According to the experimental results, the authors show how the BTT model can reduce bias and improve the accuracy of preference modeling.

**Strengths:**

1. This paper proposes the BTT model in RLHF for more reliable preference learning.
2. This paper provide the theoretical results for the readers to better understand the effectiveness of the proposed method.
3. This paper is well written and easy to be understanded.

**Weaknesses:**

1. **Method:** This paper based on the problem setting of the existence of ties, but the problem setting needs to relabel the preference data for the debias method, which is not suitable for most existing public preference datasets. For this problem, I think the author should focus on the noisy label of the preference dataset, rather than relabeling the data and make some analysis.
2. **Theory:** The theoretical analysis is based on the ground truth label of ties, rather than the noisy label modeling with the ties. Thus, the superiority of the theoretical results are not meaningful.
3. **Experiments:** The experimental evaluation results are not rational. The main simulation dataset construction is not clear enough, the evaluation on the HH dataset is also not convincing due to the lack of the ties samples. Additionally the  evaluation metrics should include the commonly used benchmark, such as alpaca eval, mt bench and arena hard etc. Also the authors should investigate some robust preference learning methods as their baselines, such as [1][2].

**References**

[1] Robust Preference Optimization for Large Language Models

[2] Robust Preference Optimization through Reward Model Distillation

**Questions:**

See weaknesses.

---

### Meta-Review · Area_Chair_aGdv · 2024-12-21

**Metareview:**

This paper explores the critical role of addressing ties in Reinforcement Learning with Human Feedback (RLHF). The traditional Bradley-Terry (BT) model for preference modeling assumes a strict preference hierarchy, overlooking the possibility that humans may view certain options as equally preferable. To overcome this limitation, the authors propose an extended Bradley-Terry model (BTT) that incorporates ties, providing a more accurate representation of human preferences. Their findings reveal that when ties are inherent in the preference model but are absent from the dataset annotations, it introduces bias into preference measurements. Experimental results demonstrate that the BTT model effectively reduces this bias and enhances the accuracy of preference modeling.

Major positive points：
+ The paper is well written.
+ The paper provides comprehensive experiments.

Major negative points
- Parts of the motivations are not well justified (reviewer k4NC)
- The experiments are not comprehensive enough to convince the reviewers.
- The simulation dataset is not well explained.

**Additional Comments On Reviewer Discussion:**

In the rebuttal period, the authors do not provide any feedback on the comments from the reviewers. Thus, all the concerns of the reviewers are not addressed, which leads to the rejection of this paper.

---

### Decision · Program_Chairs · 2025-01-22

Reject